# Classification-Based Evaluation of Multi-Ingredient Dish Using Graphene-Modified Interdigital Electrodes

**DOI:** 10.3390/mi14081624

**Published:** 2023-08-17

**Authors:** Hongwu Zhu, Yongyuan Xu, Shengkai Liu, Xuchun He, Ning Ding

**Affiliations:** Shenzhen Institute of Artificial Intelligence and Robotics for Society (AIRS), Shenzhen 518172, China; zhuhongwu0918@gmail.com (H.Z.); xuyongyuan@cuhk.edu.cn (Y.X.); liushengkai@cuhk.edu.cn (S.L.)

**Keywords:** electronic tongue, principal component analysis, electrochemical impedance spectroscopy, taste sensors, graphene

## Abstract

A taste sensor with global selectivity can be used to discriminate taste of foods and provide evaluations. Interfaces that could interact with broad food ingredients are beneficial for data collection. Here, we prepared electrochemically reduced graphene oxide (ERGO)-modified interdigital electrodes. The interfaces of modified electrodes showed good sensitivity towards cooking condiments in mixed multi-ingredients solutions under electrochemical impedance spectroscopy (EIS). A database of EIS of cooking condiments was established. Based on the principal component analysis (PCA), subsets of three taste dimensions were classified, which could distinguish an unknown dish from a standard dish. Further, we demonstrated the effectiveness of the electrodes on a typical dish of scrambled eggs with tomato. Our kind of electronic tongue did not measure the quantitation of each ingredient, instead relying on the database and classification algorithm. This method is facile and offers a universal approach to simultaneously identifying multiple ingredients.

## 1. Introduction

For culinary arts, humans hold a great advantage over automated robotic systems. Evaluation for dishes still need humans to judge tastes. Tastes involve complex overlaid processes, such as bitterness and other flavors, which are hard to identify using selective electrodes [1,2,3]. The automation evaluation of dishes is highly in demand and challenging for researchers. Although tastes remain a stubbornly subjective experience that depend on personal preferences, automatic dish evaluation systems, such as taste sensors and electronic tongue techniques, have been developed to distinguish dish samples [4,5].

Human tongues contain hundreds of taste buds capable of identifying complex tastes [6,7]. Researchers generally believe at least five tastes to exist: saltiness, sweetness, sourness, umami, and bitterness [5]. Given that taste is fundamentally a chemical process, its automation is potentially viable [8]. Selective electrodes have been used to measure changes in potential or current to identify the concentrations of certain species [9,10,11]. For example, sourness is measured via pH meter. Saltiness is measured with sodium-ion-selective electrodes. Umami is detected using a receptor for glutamate. Although different types of sensors to quantify the specified quantitative concentrations to mimic human tongues have been developed [12,13,14], the exact number of tastes is almost imperceptible. And tastes involve complex overlaid processes, such as bitterness and other flavors [5], meaning it is very challenging to develop a tool to detect all tastes [15].

Meantime, electronic tongues work in a different way. Rather than giving exact concentrations of individual species, electronic tongue tends to combine signals from non-specific sensors with pattern recognition routines [15,16,17]. Various techniques have been adopted to acquire information, such as potentiometry [18], voltammetry [19,20,21], surface acoustic waves [22], or optical chemistry [23]. In the collected data, structure and correlation can be built using multivariate methods or artificial neural nets (ANN) [16,24]. Models of calibration data could be used to predict the test dish and present further evaluation. All these algorithm require the interfaces of sensors sensitive to a broader gustatory substance, including inorganic ions and organic compounds. Graphene has been developed and investigated widely for surface modification of sensors [25,26,27]. Graphene, with its unique one-atom sheet structure, has intriguing properties, such as good electrical conductivity, high Young’s modulus, and stability. Every carbon atom in graphene is a surface atom, ensuring its high sensitivity to adsorbed molecular species [28]. Unlike metal surfaces that tend to become corroded, graphene shows more stable physicochemical properties. And graphene has been widely applied in anti-corrosive coatings for metal matrices [29]. Meantime, by controlling the ratio of hydrophilic functional groups in graphene, the degree of graphitization can be adjusted [25]. Graphene can show broad hydrophilicity with a large number of polar and non-polar molecules [30], providing a good candidate for sensor surface modifications.

Herein, we prepared an interdigital electrode with a modified graphene layer and develop a method based on a machine learning algorithm to evaluate mixed systems, such as stir-fried dishes. The interdigital electrode (IE) with Cu/Ni/Au was prepared by a commonly used nano-fabrication technology. Subsequently, a layer of graphene was electrodeposited from graphene oxide (GO) to modify the surface physicochemical properties of bare gold metal [31,32]. The electrochemically reduced graphene oxide (ERGO) was assembled parallel to the metal plane in a multilayer stacked structure. The as-prepared modified electrode showed good signal acquisition capability in mixed solutions. We established a database of electrochemical impedance spectroscopy (EIS) by varying the desired variables [33]. EIS data covers a large amount of information about concentrations of individual species, and is a powerful tool for investigating multicomponent systems in solutions and solids [34,35]. We analyzed the EIS data using multivariate data analysis and principal component analysis (PCA) techniques [27,36]. Suitable management and prediction models were built. A mixed solution consisting of typical cooking condiments was used as a model for verification. Further, we demonstrated this method on a typical dish of scrambled eggs with tomato. Compared with traditional electronic tongue systems, the main contributions of this study lie in the following:(i)The ERGO showed high graphitization and good conductivity, which facilitates the electron transfer between the interface of ions in solution and the electrode and the interface of graphene and gold.(ii)The sensing system could successfully evaluate the deviations from a standard of a sample without quantitative analysis. Our electronic tongue is facile and offers a more universal approach to simultaneously identifying multiple ingredients.

## 2. Materials and Methods

### 2.1. Materials and Characterization

The IE electrodes consisting of Cu/Ni/Au were commercially available from Changchun Beirun Electronic Technology Co., Ltd. (Changchun, China). GO was prepared with a modified Hummer’s method [37]. Other reagents were purchased from Sinopharm Chemical Reagent Co., Ltd. (Shanghai, China), and were used directly without any further purification. Scanning electron microscopy (SEM) images were obtained from a Zesiss Supra 40 scanning electron microscope at an accelerating voltage of 5 kV. Transmission electron microscopy (TEM) was performed on H-7650 (Hitachi, Tokyo, Japan) at an acceleration voltage of 100 kV. Powder X-ray power diffraction (PXRD) analyses were carried out on a Philips X’Pert PRO SUPER X-ray diffractometer equipped with graphite-monochromatized Cu Kα radiation. Raman scattering spectra was conducted on a Renishaw System 2000 spectrometer using the 514.5 nm line of Ar+ for excitation. All the electrochemical characterization was performed on a CHI 660E electrochemical workstation. Electrochemical impedance spectroscopy (EIS) was performed under a frequency range of 0.1 Hz to 100 kHz with an alternate current amplitude of 5 mV and an initial potential of open circuit potential.

### 2.2. Preparation of ERGO Modified IE

A 5 mg mL^−1^ GO aqueous dispersion was diluted with buffer solution to produce a concentration of 1 mg mL^−1^ GO dispersion, which was then used as an electrolyte for the electrochemical deposition. The commercial buffer solution was a mixture of NaCl (0.4 g L^−1^), KCl (0.4 g L^−1^), CaCl_2_·H_2_O (0.795 g L^−1^), urea (1 g L^−1^), Na_2_S·H_2_O (0.005 g L^−1^), and NaH_2_PO_4_·H_2_O (0.78 g L^−1^) according to ISO/TR10271 standard [38]. IE was thoroughly washed under ultrasonic with acetone, alcohol, and deionized water (DIW) sequentially. In the three-electrode system, the saturated calomel electrode (SCE) was used as the reference electrode, a graphite rod as the counter electrode, and IE as the working electrode. The distance between the electrodes was about 1 cm. A direct current voltage of −1 V was applied for 300 s. After electrodeposition, the IE was gently picked out and immersed in 100 mL DIW for 10 min. Then, it was gently rinsed by DIW and alcohol, and dried naturally.

### 2.3. Method for Database Establishment for Dish Evaluation

Modeled dishes: A mixed solution of 513 mmol L^−1^ sodium chloride (NaCl), 147 mmol L^−1^ sucrose, and 175 mmol L^−1^ monosodium L-glutamate (MSG) was prepared according to the sauce of certain Chinese stir-fried dishes. The above mixed solution was used as the standard solution. Then, concentrated solutions were prepared. Typically, 6 g NaCl was added into 20 mL DIW; 32.624 g sucrose was added into 20 mL DIW; 11.84 g MSG was added into 20 mL DIW. Afterwards, 245 μL concentrated NaCl solution was added into 25 mL standard solution each time for 20 times. After each addition, the corresponding EIS was recorded, and the electrode was carefully rinsed and dried. Similarly, the addition of concentrated sucrose and MSG at each time were 260 μL and 360 μL. A total of 400 μL DIW was added each time into standard solution, 14 times, to generate the diluted samples. All the EIS data ranged from 0.1 Hz to 100 kHz using 20 pairs of IEs modified with graphene at room temperature.

Dish of scrambled eggs with tomato: A sample database of nine dishes of scrambled eggs with tomato was established using the same method above. A volume of 5 mL soup of commercial scrambled eggs with tomato was diluted to 25 mL, and filtrated to use as the standard sample. Then, the above concentrated NaCl (245 μL) solutions were added into the standard sample, and the corresponding EIS was recorded. We repeated the addition another two times, and three salty samples were obtained. After each test, the electrode was carefully rinsed with DIW and ethanol. Similarly, another two groups of sucrose group and MSG group samples were prepared. Each addition of sucrose was 260 μL, and those of MSG were 360 μL. Thus, a total of nine dish samples were collected for the PCA classification.

## 3. Results and Discussions

(i)Modification of IE electrodes with ERGO

As illustrated in Figure 1, the metal layer of the commercial IE of 20 pairs fingers consisted of Cu/Ni/Au with thickness of 12 μm, 1 μm, and 1 μm from bottom to top layers. The Cu layer provided the conductivity, the Ni layer was deposited for adhesion, and the Au layer was used to provide stable chemical interfaces. The line width and distance were all 100 μm. The IE provided a high specific surface area for capacitive sensing. The operating principle of the IE resembles stacked parallel-plate capacitors. GO was electrochemical reduced on the IE for modification. The electrodeposition method was facile and scalable. The layer thickness could also be regulated via variation of deposition time. The buffer solution was used to control the pH to suppress the gas generation. To obtain a uniform layer, the IE should be carefully cleaned. Unlike common reports for single component detection, we used a model of mixture solution containing three variables: NaCl, sucrose, and MSG by increasing the concentration of each variable to construct the vector of dish detection. A set of samples were used to build the database, which was used for unknown sample prediction and evaluation.

As shown in Figure 2a, GO prepared with Hummer’s method, a commonly used chemical exfoliation method, shows ultrathin nanosheet morphology. The lateral size could reach 10 μm. Wrinkles appeared on the nanosheets due to the high interaction forces after water evaporation. Under the electric field during electrodeposition, the nanosheets were parallelly attached on the negative electrode and were reduced simultaneously. Other components could also be added into the solution to produce composites, such as poly(3,4-ethylenedioxythiophene) or metal precursors. We chose unmodified ERGO to ensure the robustness of the electrodes. Powder X-ray diffraction (PXRD) patterns of ERGO are shown in Figure 2b. The featured diffraction peak of GO at 10.3° disappeared, and a new peak near 26° formed, indicating the recovery of graphitic crystals. The diffraction peak was broad, owing to the stacking of ERGO nanosheets. Raman spectra showed the degree of reduction of GO. Two prominent peaks centered at 1346 cm^−1^ and 1587 cm^−1^ could be observed, corresponding to the D band and G band, respectively. The D band resulted from the defects or structural disorder, and the G band was in accordance with the E2g mode of sp2 hybridized carbon atoms. The intensity ratio of the G to D band (I_G_/I_D_) for ERGO was 0.934, which was higher than that of pristine graphene (0.714). This suggests defects on the GO sheets were removed after electrochemical reduction. The electrodeposition at different potentials were also investigated: I_G_/I_D_ were 0.76, 0.659, 0.665, for −1.1 V, −1.3 V, and −1.5 V, respectively. In addition, the 2D, 2D’ and the combination mode D + D’ were shown in this spectrum around 2800 cm^−1^, indicating the existence of both monolayers and multilayers of ERGO nanosheets.

The deposition of the ERGO layer also changed the wettability of the electrode surface (Figure 2d). The Cu/Ni/Au surface had a water contact angle (CA) of 73.8° in the air. The Cu/Ni/Au/ERGO surface had a CA of 87.9° because the graphene modification lowered the surface free surface energy. The hydrophobicity helped keep the electrode clean and increase the compatibility with nonpolar molecules. As shown in Figure 2e, the nanosheet formed a dense, multilayer, stacked structure. The red dashed lines are the edges of the nanosheets. The interaction between layers was the strong van der Waals interaction, ensuring the mechanical robustness and electron transportation. The transmission electron microscopy (TEM) image in Figure 2f also reveals the detailed structure of the ERGO layers. Samples were scraped off and dispersed under ultrasonic. The nanosheets were much thicker compared with that in Figure 2a, and the material possessed a thin lamellar structure with distinct edges, overlap, wrinkles, and curved profiles.

(ii)Electrochemical characterization of individual ingredient

Cell resistance was directly tested with a constant potential of 0.1 V. The cell resistances were about 50 kΩ for both 1 mg mL^−1^ and 5 mg mL^−1^ aqueous GO solution. EIS curves of the two aqueous GO solution were also employed to study the deposition process in Figure 3a,b. The Bode curves of phase in Figure 3b showed that the aqueous GO solution of high content had higher sensitivity to the frequency. For the Nyquist curves, the intersection with the *x*-axis at high frequencies was attributed to the equivalent series resistance (R_s_), including aqueous GO solution, the contact resistance at the electrode–GO interface, electrode leads, and terminals. The corresponding resistances were 1326 Ω and 274 Ω for 1 mg mL^−1^ and 5 mg mL^−1^ GO solutions. Furthermore, a partial semicircle was formed in the high frequency region, the diameter of which could be used to calculate the charge transfer resistance of the electrode (R_ct_). The straight line at the low frequencies stands for the diffusion behavior of aqueous GO at the interfaces. The steeper sloped line represents an ideal capacitive behavior with the faster diffusion of ions in electrolyte. The 5 mg mL^−1^ GO solution exhibited lower R_s_ and R_ct_ than 1 mg mL^−1^ GO. And the ions’ diffusion behavior was almost the same, which was mainly affected by the temperature.

The EIS curves of NaCl, sucrose, and MSG at concentrations of 1 mM, 10 mM, 100 mM, and 1000 mM were recorded, respectively. As shown in Figure 3c–e, the Nyquist curves showed distinct differences. The equivalent series resistances, R_s,_ are summarized in Table 1. All the conductivity was decreased as the concentration increased. At certain concentration, the order of conductivity was MSG, NaCl, and sucrose. Even though sucrose is a molecular crystal without conductivity, EIS could provide the R_s_ results, which might be a result of the interaction at the interfaces due to the molecular polarity. As shown in Figure 3f, no obvious peak of sucrose was observed. The peak of MSG was broader than NaCl. And the peak gradually shifts towards the lower frequencies. Due to NaCl and MSG being typical ion crystals, their R_s_ were similar at the same concertation. Thus, the phase might be the feature for discrimination. Sucrose is a kind of nonelectrolytic molecular crystals of polyhydroxyl compound. Most of the flavoring substances in dishes are non-electrolyte; effective interfacial interactions were crucial for detecting such substances. Molecular polarity and the formation of double-layer capacitance at the electrode interfaces are the available information. The graphical shapes including the values of R_s_ was quite different from the others. The sucrose solution also responded to the variable frequency AC signals, attributed to the surface modification of ERGO on the metal electrodes.

(iii)Classification-based evaluation in dimensions of cooking seasoning

Detection of a mixed solution was more challenging and meaningful than that of single components. The mixed solution of 513 mM NaCl, 147 mM sucrose, and 175 mM MSG was used as the model for dishes. Then, a salty samples set, sweeter samples set, and high-MSG samples set were built following the methods in the experimental section. The data set was classified via multivariable analysis and was use to evaluate the unknown sample of the model dish. The raw data of the 3D plot was present in Figure 4a. A total of 73 parameters were obtained from one time sampling scan. Resistance (Z’) decreased and the reactance (Z″) became less negative with increasing frequency. Curves were close together at high frequencies, and split up gradually as the frequency decreased. As shown in Figure 4b, three main groups could be seen at the enlarged view of the high frequencies region, where the lowest resistance group contains salty samples, the second high-MSG samples, and sweeter sample in the third. As shown in Figure 4c, the phase sweeter samples overlapped, resulting from the coverage of salt in the mixture. The peak shifts of MSG were still preserved.

A quantitative interpretation of EIS was not straightforward. Multivariate data analysis and principal component analysis (PCA) were used to handle and visualize the acquired data. PCA projects the data in reduced dimensions, defined as the principal components (PCs). This technique was useful for identifying patterns without supervision, when there were correlations present among the data. The raw data were standardized to eliminate influence of dimension, i.e., each variable was mean centered and divided by its standard deviation to become “equally important”. If the number of available samples were few, a leave-one-out cross-validation procedure could be used to build and validate the models. Figure 4d demonstrates the score plot of PCA for three preprocessed groups describing the projection of samples defined by the first (PC1) and second (PC2) components. PC1 accounted for the most variation in impedance values (82.8%), while PC2 accounted for the next largest variation (12.1%). Based on the score plot, three groups could be separated along PC1 and could be distinguished clearly.

We also set up a classification task to evaluate an actual dish of scrambled eggs with tomato, a popular dish worldwide, as shown in Figure 4a. In our application, we used previous condiments as taste metrics. NaCl corresponds to saltiness, sucrose corresponds to sweetness, and MSG corresponds to delicate flavor. This was reasonable because the condiments are the controllable variable for cooks. Three subsets with each of three samples were then used to train and validate the PCA classifier. The EIS curves of the nine samples are shown in Figure 4e. Each curve contain 73 features from 0.01 Hz to 100 kHz. Despite the physical meaning of electrochemistry, distinguishing the subsets intuitively was almost impossible. Figure 4f shows the score plot after PCA for three subsets along the PC1 and PC2. PC1 accounted for the most variation in impedance values (76.4%), while PC2 accounted for the next largest variation (23.6%). Other classification algorithm such as K-nearest neighbors, decision trees, and support vector machines were also effective in carrying out such tasks [39]. The distances of samples in sucrose subsets were closer due to the larger molecular volume, which were difficult to adsorb and desorb under the AC current disturbance of EIS. Overall, the three subsets could be distinguished clearly using our electrodes, giving an evaluation of the practical dish.

## 4. Conclusions

In summary, we developed a preliminary method for mixed solution classification, which could be potentially used in dish evaluation. Firstly, a mixed solution of cooking seasoning was used to simulate a Chinese stir-fried dish. By adding a series of components, we built a group set for sample prediction. The sensing electrode was prepared via depositing graphene on the interfaces, ensuring the effective acquisition of the solution information. EIS was also conducted at a wide frequency range. By using multivariate data analysis to evaluate the parameters from the measurements, we could predict the quality of samples in mixture systems. Further, we demonstrated the effectiveness of this method on scrambled eggs with tomato. Three subsets were classified with PCA to show which cooking seasoning had been added. We believe this work might have potential application in evaluation of dishes or environmental monitoring of mixed solution systems.

## Figures and Tables

**Figure 1 micromachines-14-01624-f001:**
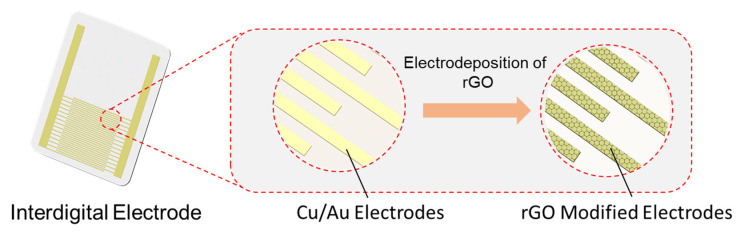
Schematic diagram of the graphene-modified interdigital electrode for electronic tongue. The surface of Cu/Ni/Au IE was covered with a layer of rGO via electrodepositing in GO solution.

**Figure 2 micromachines-14-01624-f002:**
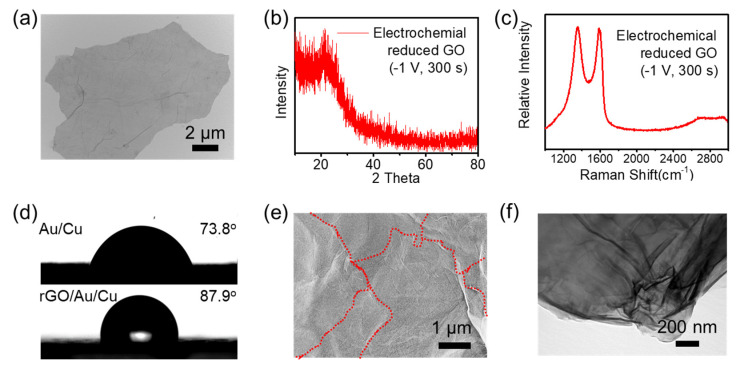
Characterization of the deposited ERGO. (**a**) The GO nanosheets prepared via modified Hummer’s method. (**b**,**c**) The PXRD and Raman spectrum of ERGO to identify the graphitization. (**d**) The ERGO modification of the electrode changed the wettability of the surface. (**e**,**f**) The ERGO nanosheets were multilayer stacked, and were parallelly deposited on the electrode surface. The red dashed lines are the edges of the nanosheets.

**Figure 3 micromachines-14-01624-f003:**
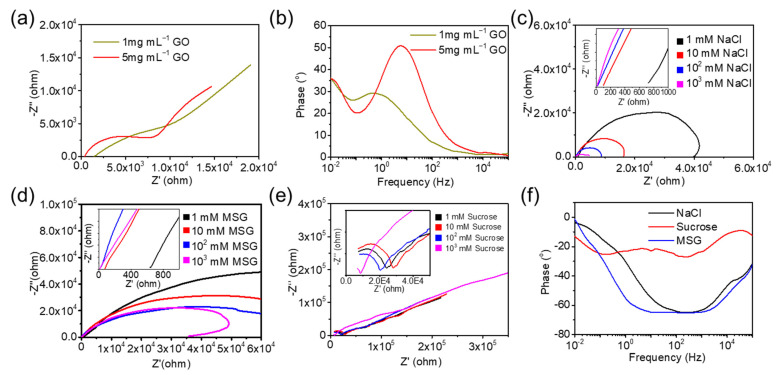
The Nyquist curves (**a**) and the Bode curves (**b**) of GO solution of 1 mg mL^−1^ and 5 mg mL^−1^ used for ERGO deposition. (**c**–**f**) The Nyquist curves and the phase variations of different species with varied concentration versus frequency response.

**Figure 4 micromachines-14-01624-f004:**
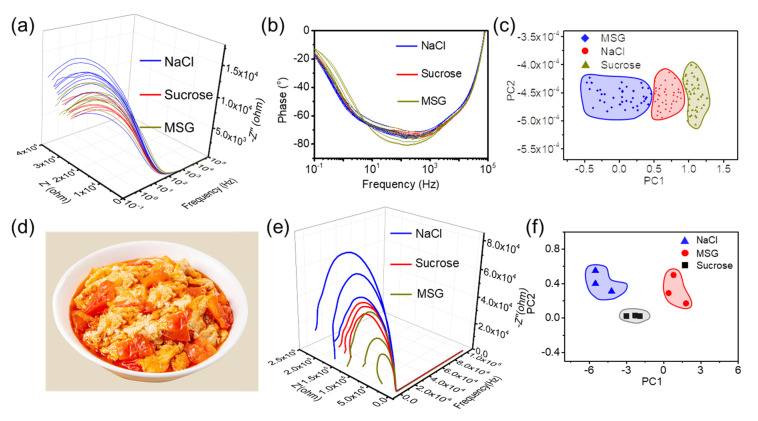
(**a**) The 3D EIS of model solution with certain variable ranging from 0.1 Hz to 1 × 10^5^ Hz. (**b**) The enlarged view of the high frequency region (**c**) The result for evaluating standard solution made with impedance data and PCA modelling. (**d**) The typical dish of scrambled eggs with tomato was used for demonstration. (**e**) The EIS curves of the raw data. (**f**) The result after PCA clearly show the classification of the sample subsets, which could be referred to assist the cooking.

**Table 1 micromachines-14-01624-t001:** Rs values in EIS curves of the samples in three subsets.

Categories	Concentrations of Subsets			
	1 mM (Ω)	10 mM (Ω)	100 mM (Ω)	1000 mM (Ω)
NaCl	688.5	91	25	7.5
MSG	631.1	71.7	7.4	1.8
Sucrose	23,278	19,875	16,434	779

## Data Availability

Not applicable.

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
