# Peer review of "Classification-Based Evaluation of Multi-Ingredient Dish Using Graphene-Modified Interdigital Electrodes"

_micromachines, 2023, doi:10.3390/mi14081624_

Round 1
Reviewer 1 Report
In the manuscript, a sensor based on Graphene modified gold interdigital electrode is used to realize the classification and identification of multi-ingredient dishes. Hope the author can make adjustments to the content of the manuscript. Here are some reviews:
Q1. In the introduction of the manuscript, the shortcomings of electronic tongue in detection that cannot provide accurate concentration are pointed out. However, the quantitative prediction results of the prepared sensor for the taste were not provided in the manuscript. What are the advantages of this sensor compared to electronic tongue?
Q2. The explanation in the manuscript on how to use multivariate data combined with principal component analysis (PCA) to analyze impedance spectrum data and effectively distinguish the three flavors in the dish model is not clear enough. In Figure 4, the distinction is not obvious.
Q3. The manuscript only analyzed the mixed solution model configured by oneself, but lacked research on dish samples. Can you increase the analysis results of actual samples to demonstrate the effectiveness of sensors in practical applications?
Q4. The images in the manuscript are not clear and many curves are difficult to distinguish. Please adjust them.

English is acceptable, some words need to be revised
Author Response
We appreciate the comments from all two referees, helping us improve the quality. For the purpose of clarity, the point-to-point answers are marked with RED color and started with “**” below. And please see the attachment of answers with Figures.
In the manuscript, a sensor based on Graphene modified gold interdigital electrode is used to realize the classification and identification of multi-ingredient dishes. Hope the author can make adjustments to the content of the manuscript. Here are some reviews:
Q1. In the introduction of the manuscript, the shortcomings of electronic tongue in detection that cannot provide accurate concentration are pointed out. However, the quantitative prediction results of the prepared sensor for the taste were not provided in the manuscript. What are the advantages of this sensor compared to electronic tongue?
** Thanks for this valuable comment. One of the typical electronic tongue are also called the ion selective electrodes. They are capable of detecting concentrations. However, the flavor of the real dishes is very complex including not only the ions but also the other flavor substances of organic compounds. So it is almost impossible to evaluate the tastes of one dish by quantifying of every component that affects taste perception.
In this concern, we take another different strategy of big data methods. We set the well-known dish as the standard, and try to evaluate the deviations of another given unknown sample, which is different new method. There is no need to know the quantitative prediction results of specified quantitative concentrations. We still provide evaluations of simulated dish from several dimensions. The sensor was required to be capable of obtaining more overall information through the interface actions, and the graphene modification deposited on the surface is crucial.
The advantages of our sensor show higher activity over conventional metal electrodes due to the graphene layer. Combining our sensor with electronic impedance spectrum technique, our electronic tongue is a more universal approach for different dishes. And the sensitivity meets requirements of human taste perception. After all, the sensitivity of taste depends on the human individual.
According to this suggestion, this manuscript was further revised as follows. Section 1, paragraph 2, was reorganized to stress on our point. In Section 1 paragraph 4, the sentence has been added by points:
“Compared with traditional electronic tongues system, the main contributions of this letter lie in the following twofold:
(i) The ERGO showed high graphitization and good conductivity, which facilitates the electron transfer between the interface of ions in solution and electrode and the interface of graphene and gold.
(ii) The sensing system could successfully evaluate the deviations from standard of a sample without quantitative analysis. Our electronic tongue is a facile, and more universal approach to simultaneously identifying multi ingredients.”
Q2. The explanation in the manuscript on how to use multivariate data combined with principal component analysis (PCA) to analyze impedance spectrum data and effectively distinguish the three flavors in the dish model is not clear enough. In Figure 4, the distinction is not obvious.
** We thank the reviewer’s good comments here. PCA algorithm is one of the machine learning algorithms. It is a method for data dimension reduction and feature selection, used to simplify data sets and retain the most important aspects of the data. PCA is widely used in machine learning for tasks such as data compression, feature extraction, and classification.
To distinguish the three flavors in the dish model, we use the following steps including data collection, data exploration and cleaning, PCA deployment, and visualization. (a) Data collection includes the design of the standard sample containing three flavors, the deviated samples of each three dimensions by adding or certain components to build the data base. (b) Data exploration and cleaning includes the check of the raw data and the outlier dection to determine to recollect the curve one more time. (c) PCA deployment used the build-in toolbox function in Matlab. (d) Visualize the data after dimensionality reduction by PCA into a graph to obtain the results. We has clearly point out these steps in the Section 2. Materials and Methods and Section 3. Results and Discussions in this revision. Typical process can also referred to citations (Sairin et al., International Conference on Sensing Technology (ICST), 739; Ulrich et al., Sensors & Actuators B Chemical, 2007, 127, 613-618; Facure et al., Talanta, 2017, 167, 59-66.)
Figure 4 was revised to give a clear distinction between the different categories. Samples near the borders are the samples close to the standard sample in our data base. In practice, these samples are only just a little salty or so compared with the standard sample, which is reasonable.
Q3. The manuscript only analyzed the mixed solution model configured by oneself, but lacked research on dish samples. Can you increase the analysis results of actual samples to demonstrate the effectiveness of sensors in practical applications?
** Thanks for this valuable comment. Dish of scrambled eggs with tomato is typical actual samples of multi-ingredient in practical applications (Sochacki et al., Frontiers in Robotics and AI, 2022, 9.). Here, we took this dish as an actual sample to demonstrate the effectiveness of sensors in practical applications.
In revision, the experimental process was added in the Section 2, 2.3 Method for database establishment for dish evaluation: “Dish of scrambled eggs with tomato: A sample database of nine dishes of scram-bled eggs with tomato was established using the same method above. 5 mL soup of a commercial scrambled eggs with tomato was diluted to 25 mL, and filtrated to use as the standard sample. Then, the above concentrated NaCl (245 μL) solutions was added into the standard sample, and the corresponding EIS was recorded. Repeated the addition for another two times, three salty samples were obtained. After each test, the electrode was carefully rinsed with DIW and ethanol. Similarly, another two group of sugar group and MSG group samples were prepared. Each addition of sugar was 260 μL, and MSG was 360 μL. Thus, totally nine dish samples were collected for the PCA classification.”
And the results and discussions were supplemented in Fig. 4(d-f) as follows.
“We also set up a classification task to evaluate an actual dish of scrambled eggs with tomato, a popular dishes worldwide as shown in Fig. 4a. In our application, we use previously condiments as taste metrics. NaCl corresponds to saltiness, sugar cor-responds to sweetness, and MSG corresponds to delicate flavor. This is reasonable be-cause the condiments were the variable for cookers that could be controllable. Three subsets with each of three samples are then used to train and validate the PCA classifier. The EIS curves of the nine samples was in Fig. 4e. Each curve contain 85 features from 0.01 Hz to 100 kHz. Despite the physical meaning of electrochemistry, distinguish the subsets intuitively is almost impossible. Fig. 4f showed the score plot after PCA for three subsets along the PC1 and PC2. PC1 accounted for the most variation in impedance values (76.4 %), while PC2 accounted for the next largest variation (23.6 %). Other classification algorithm such as K-nearest neighbors, decision trees, and support vector machines were also effective to carry out such tasks [37]. Distance of samples in sugar subsets were closer due to the larger molecular volume are difficult to adsorb and de-sorb under the AC current disturbance of EIS. Overall, the three subset could be distinguished clearly using our electrodes, giving an evaluation of the practical dish.”
Q4. The images in the manuscript are not clear and many curves are difficult to distinguish.
Please adjust them.
** We thank the valuable comment here. The Font size of the legends, axis’s labels, and curves are revised to give a clear show of the data. Please refer to all the figures in this revision.

Reviewer 2 Report
The paper presented for review presents the modification of gold interdigitated electrodes using electrochemically reduced GO. The electrodes were used to study solutions containing NaCl, sugar (likely sucrose), and monosodium L-glutamate using EIS (Electrochemical Impedance Spectroscopy) technique. The aim of the work was to develop an electronic tongue for taste recognition. The research can be considered preliminary and potentially promising; however, due to the chaotic manner in presenting the research results and a lack of many basic pieces of information, it is difficult to definitively assess the merits of the work. Detailed comments have been compiled in the form of a bulleted list below.
1. There is no information in the title on the subject of the research.
2. The first four sentences of the abstract are suitable for the Introduction. The abstract needs to be rewritten because there is no information about what was studied. The abstract should include information about the type of electrode used (gold IE electrodes, modified by electrochemically reduced graphene oxide), what media were examined (synthetic solution simulating the composition of spices used in the preparation of stir-fried dishes) and what was the aim of an this work (construction of artificial sensing system employing the EIS technique capable of differentiation different tastes).
3. ‘The AI robotic systems require not only cooking system but also auto evaluation systems for dishes to replace human tasters’ - The sentence needs to be rewritten. What do 'AI robotic systems' refer to?
4. ‘Taste is hopeful to realize automation because it is fundamentally a chemical process’ - The sentence needs to be changed. An example of a revised version is: 'Given that taste is fundamentally a chemical process, there is potential for its automation’.
5. All these require sensors to be able to collect a broader information, which traditional metal electrode couldn’t meet the demands’ – The sentence needs to be rewritten.
6. ‘Unlike metal surfaces that tend to bind with sulfur compounds, graphene showed a stable physicochemical properties and broad compatibility with a large number of species’ - Please expand on this section. It is a justification for why the authors decided to use not just gold electrodes but gold electrodes modified with rGO (reduced graphene oxide). Please add literary references to works dedicated to methods of modifying gold electrodes with rGO.
7. Please replace 'stir-fired' with 'stir-fried', 'stir-frying', or their synonyms.
8. ‘The interdigital electrode (IE) with Cu/Ni/Au was prepared by a commonly used nano-fabrication technology’ - In the text, there is information that these were commercially available electrodes. Please add information on the manufacturer of the electrodes. What function did Cu (copper) and Ni (nickel) serve? Were they auxiliary layers, improving adhesion, or were they electrodes with a multicomponent (multielement) surface? Later in the manuscript, information about nickel in the composition of the electrodes disappears. What type of electrodes did the authors work with: Cu/Ni/Au, Cu/Au, or 'bare gold metal'? If Ni and Cu served only an adhesive function, it will suffice to add this information during the description of the devices used in the work.
9. ‘A mixture of solution was used as a model of Chinese stir-fired dish’ - This information is definitely exaggerated. The authors prepared a mixture of NaCl, sugar (sucrose?), and monosodium L-glutamate. A solution with such a composition can only be treated as a rough approximation of the spices used during stir-frying.
10. ‘The commercial IE consisted of Cu/Ni/Au with thickness of 12 μm, 1 μm, and 1 μm’ – There is no information about the electrode manufacturer. Do the details pertain to the thickness of the Cu (12 µm), Ni (1 µm), and Au (1 µm) layers? What was the composition of the electrode surface that was in contact with examined solution? Was it pure gold?
11. ‘5 mg mL-1 GO was’ - Was it a water suspension?
12. ‘GO solution was the electrolyte’, ‘And buffer solution was used to control the pH in case of gas generation’ - Please organise the information about the solutions used, namely, provide details about the composition of the solution used for depositing GO on the IE, including the concentration of GO, e.g. the optimal one. Was the IE electrode rinsed after depositing GO? If so, please provide information on what was used to rinse the electrode. Was more than one buffer used during the deposition of GO on the IE surface? What was the pH of the solution used to deposit GO, described in the manuscript as 'The buffer solution was a mixture of NaCl (0.4 g L-1), KCl (0.4 g L-1), CaCl2·H2O (0.795 g L-1), urea (1 g L-1), Na2S·H2O (0.005 g L-1), NaH2PO4·H2O (0.78 g L-1)'? Was the composition of the solution taken from the literature, or is it the result of the authors' research? Please provide literary references. At what potential was the hydrogen evolution observed? Did the hydrogen evolution interfere with the deposition of GO?
13. ‘2.3 Evaluation method of modelled dish’ - The entire chapter needs to be revised, including the title of the chapter. Provide information about the composition of the standard solutions and the solution used to record the EIS curves. 'Afterwards, 245 μL of NaCl solution was added into the model dish each time, for 20 times. After each addition, the corresponding EIS was recorded.' – What was the purpose of these studies? Was the EIS response recorded to obtain a calibration curve? The results for the 20 NaCl additions are not presented in the work.
14. Results and Discussions The chapter should be divided into sub-sections: (i) Modification of Au electrodes with rGO (possible/optional subtitle for the subsection). In this chapter, microscopic studies, XRD, Raman, and CA should be included. To allow the reader to assess the changes caused by the reduction of GO to rGO, please add the XRD curves recorded for GO before electroreduction. What was the reduction potential used during the rGO study shown in Figures 2b and 2c? Can the authors add to the figure the Raman spectra of rGO obtained after reduction at -1.1 V, -1.3 V, and -1.5 V? Did the wetting angle measured for the modified IE/rGO electrode surfaces depend on the reduction potential of GO? What reduction potential was used to prepare the electrodes for which CA was measured, as seen in Figure 2d? What was the reduction potential used to create the rGO visible in Figures 2e-2f? Regarding 'Samples were scraped off and dispersed under ultrasonic' – How were the rGO samples prepared for XRD and Raman studies? At the end of the chapter, information about the optimal conditions for preparation of rGO/Au electrodes should be included, considering the buffer composition, pH, GO concentration, reduction potential, and deposition time. (ii) Use of rGO/Au electrodes to study solutions that simulate the composition of spices (possible/optional subtitle for the subsection). This chapter should contain information on EIS studies.
15. ‘(a-b) GO of different concentration showed different EIS’ - This is a conclusion, change the figure description. There is no information about the conditions for preparing the electrode and the composition of the solutions studied. From the figure, it is not clear whether single-component or multicomponent solutions were examined
16. 16) ‘EIS behavior of three pure components have also been studied to demonstrate the data feature’ - Please revise; the sentence is unclear.
17. ‘the Nyquist curves showed distinct differences’ - Information about Rs for solutions of different compounds and concentrations should be placed in a table.
18. ‘We refer to the concentration certain Chinese stir-fired dishes, and diluted for 10 times as pretreatment’ - Please revise; the sentence is unclear.
19. ‘we developed an evaluation system for dishes’ - This conclusion goes too far. The authors investigated only a synthetic solution consisting of NaCl, sugar, and monosodium L-glutamate. Such a solution can at best be an approximation of the composition of a seasoning in liquid form, not a finished dish.
20. Please correct the description of Figure 1. The use of the word 'illustrator' is not appropriate.
The article requires thorough linguistic correction. This includes a detailed review of grammar, sentence structure, and clarity to ensure that the ideas and findings are conveyed accurately and coherently. Special attention should be paid to terminology specific to the field.
Author Response
We appreciate the comments from all two referees, helping us improve the quality. For the purpose of clarity, the point-to-point answers are marked with RED color and started with “**”.Please see the attachment with figures. And the answers were as follows:
Point 1: The paper presented for review presents the modification of gold interdigitated electrodes using electrochemically reduced GO. The electrodes were used to study solutions containing NaCl, sugar (likely sucrose), and monosodium L-glutamate using EIS (Electrochemical Impedance Spectroscopy) technique. The aim of the work was to develop an electronic tongue for taste recognition. The research can be considered preliminary and potentially promising; however, due to the chaotic manner in presenting the research results and a lack of many basic pieces of information, it is difficult to definitively assess the merits of the work. Detailed comments have been compiled in the form of a bulleted list below.
Response 1: ** Thanks for this valuable comment. This work is fundamental research trying to demonstrate an alternative method for the electronic tongues. We have reorganized the results presentation and supplemented some experimental results to declare merits of the work in this revision. According to this suggestion, this manuscript was also further revised by a native speaker.
Point 2: There is no information in the title on the subject of the research.In the introduction of the manuscript, the shortcomings of electronic tongue in detection that cannot provide accurate concentration are pointed out. However, the quantitative prediction results of the prepared sensor for the taste were not provided in the manuscript. What are the advantages of this sensor compared to electronic tongue?
Response 2: ** Thanks for this valuable comment. The title was revised from “Classification-based identification of multi-ingredient using graphene modified interdigital electrodes” into “Classification-based evaluation of multi-ingredient dish using graphene modified interdigital electrodes”
Graphene modified interdigital electrodes was the sensors device, facilitating the obtainment of ingredient information in solution. Classification-based declare the machine learning based PCA algorithm for the evaluation of dishes. Both sensors and algorithm method are included in this revised title.
Point 3: The first four sentences of the abstract are suitable for the Introduction. The abstract needs to be rewritten because there is no information about what was studied. The abstract should include information about the type of electrode used (gold IE electrodes, modified by electrochemically reduced graphene oxide), what media were examined (synthetic solution simulating the composition of spices used in the preparation of stir-fried dishes) and what was the aim of an this work (construction of artificial sensing system employing the EIS technique capable of differentiation different tastes).
Response 3: ** Thanks for this valuable comment. In this revision, the first four sentences of the abstract were transferred to the Introduction. And the section Abstract was rewritten as follows:
“A taste sensor with global selectivity can be used to discriminate taste of foods and provide evaluations. Interfaces that could interact with broad food ingredients are beneficial for data collection. Here, we prepared electrochemically reduced graphene oxide (ERGO) modified interdigital electrodes. The modified electrodes showed good sensitivity towards cooking condiments in mixed multi-ingredients solutions under electrochemical impedance spectroscopy (EIS). The data base of EIS of cooking condiments was established. Based on the principal component analysis (PCA), subsets of three taste dimension could be classified, which could provide the deviation of unknown dish with standard dish. Further, we demonstrated the effectiveness on a typical dish of scrambled eggs with tomato. Our kind of electronic tongue did not measure the quantitation of each ingredient, but rely on the data base and classification algorithm. This method is facile, and universal approach to simultaneously identifying multi ingredients.”
Point 4: . The AI robotic systems require not only cooking system but also auto evaluation systems for dishes to replace human tasters’ - The sentence needs to be rewritten. What do 'AI robotic systems' refer to?
Response 4: **Thanks for this valuable comments. This sentence was deleted. And this part has been revised to reduce ambiguity. Another sentence is inserted “The automation evaluation of dishes is highly in demand and challenging for researchers.”
Point 5: Taste is hopeful to realize automation because it is fundamentally a chemical process’ - The sentence needs to be changed. An example of a revised version is: 'Given that taste is fundamentally a chemical process, there is potential for its automation
Response 5: **We thank this valuable comment. This sentence was revised into “Given that taste is fundamentally a chemical process, its automation is potentially viable.”
Point 6: All these require sensors to be able to collect a broader information, which traditional metal electrode couldn’t meet the demands.’ The sentence needs to be rewritten.
Response 6: **We thank this valuable comment. This sentence was revised into “All these algorithm require the surface of sensors sensitive to a broader gustatory sub-stance, including inorganic ions and organic compounds. Graphene have been developed and investigated widely for sensors surface modification [26-28]. Graphene with unique one-atom sheet structure has intrigue properties, such as good electrical conductivity, high Young's modulus, and stability.”
Point 7: graphene showed a stable physicochemical properties and broad compatibility with a large number of species’ - Please expand on this section. It is a justification for why the authors decided to use not just gold electrodes but gold electrodes modified with rGO (reduced graphene oxide). Please add literary references to works dedicated to methods of modifying gold electrodes with rGO.
Response 7: **Thanks for this valuable comment. We add literary references (Cui, et al. A comprehensive review on graphene-based anti-corrosive coatings. Chemical Engineering Journal, 2019, 373, 104-121; Zhao et al., An overview of graphene and its derivatives reinforced metal matrix composites: Preparation, properties and applications. Carbon, 2020, 170, 302-326.), and expand to make it more abundant and clear. “Unlike metal surfaces that tend to be corroded, graphene showed more stable physi-cochemical properties. And graphene has been widely applied as anti-corrosive coat-ings for metal matrix [30]. Meantime, by controlling the ratio of hydrophilic functional groups on graphene, the degree of graphitization could be adjusted [26]. And the gra-phene could show broad hydrophilicity with a large number of polar and non-polar molecules [31], providing a good candidate for sensor surface modifications.”
Point 8: Please replace 'stir-fired' with 'stir-fried', 'stir-frying', or their synonyms.
Response 8: ** We thank the reviewer’s good comments here. All the words “stir-fired” was replaced with “stir-fried” throughout the manuscript.
Point 9: The interdigital electrode (IE) with Cu/Ni/Au was prepared by a commonly used nano-fabrication technology’ - In the text, there is information that these were commercially available electrodes. Please add information on the manufacturer of the electrodes. What function did Cu (copper) and Ni (nickel) serve? Were they auxiliary layers, improving adhesion, or were they electrodes with a multicomponent (multielement) surface? Later in the manuscript, information about nickel in the composition of the electrodes disappears. What type of electrodes did the authors work with: Cu/Ni/Au, Cu/Au, or 'bare gold metal'? If Ni and Cu served only an adhesive function, it will suffice to add this information during the description of the devices used in the work.
Response 9: ** We thank the valuable suggestion here. The interdigital electrode were commercially available and was bought directly for experiment. According to the merchants on taobao. The metal conductive electrode layer was Cu/Ni/Au prepared by thermal evaporation method, which was widely used in physical and medical sensors. Cu layer was thick to provide conductive path. Ni layer was deposited to provide adhesion layer. And Au layer was used to provide stable chemical properties.
The disappear of Ni is a typo, and was revised throughout the manuscript. The abbreviation of Cu/Ni/Au/ERGO was also revised. Ni and Cu were not participated in the sensing process, this information was added in this revision. “As illustrated in Fig. 1, the metal layer of commercial IE of 20 pair fingers were consisted of Cu/Ni/Au with thickness of 12 μm, 1 μm, and 1 μm. Cu layer provided the conductivity, Ni layer was deposited for adhesion, and Au layer used to provide stable chemical interfaces.”
Point 10: A mixture of solution was used as a model of Chinese stir-fired dish’ - This information is definitely exaggerated. The authors prepared a mixture of NaCl, sugar (sucrose?), and monosodium L-glutamate. A solution with such a composition can only be treated as a rough approximation of the spices used during stir-frying.
Response 10: ** Thanks for this valuable suggestion here. The ingredients of solution mixture used here were the common condiments during cooking. Thus, we want to prove to distinguish the qualitative analysis to guide the cooking process. Here is revision “A mixture of solution consisted of typical cooking condiments was used as a model for verification. Further, we demonstrated this method on a typical dish of scrambled eggs with tomato.”
Further, we demonstrated this method on a typical dish of scrambled eggs with tomato. The results and the discussions are supplemented as follows:
Fig. 4 (a) The 3D EIS of model solution with certain variable ranging from 0.1 Hz to 1 × 105 Hz. (b) The enlarge view at high frequency region (c) The result for evaluating modelled dishes made with impedance data and PCA modelling. (d) The typical dish of scrambled eggs with tomato was used for demonstration. (e) The EIS curves of the raw data. (f) The result after PCA clear show the classification of the sample subsets, which could be referred to assist the cooking.
We also set up a classification task to evaluate an actual dish of scrambled eggs with tomato, a popular dishes worldwide as shown in Fig. 4a. In our application, we use previously condiments as taste metrics. NaCl corresponds to saltiness, sugar cor-responds to sweetness, and MSG corresponds to delicate flavor. This is reasonable be-cause the condiments were the variable for cookers that could be controllable. Three subsets with each of three samples are then used to train and validate the PCA classi-fier. The EIS curves of the nine samples was in Fig. 4e. Each curve contain 85 features from 0.01 Hz to 100 kHz. Despite the physical meaning of electrochemistry, distinguish the subsets intuitively is almost impossible. Fig. 4f showed the score plot after PCA for three subsets along the PC1 and PC2. PC1 accounted for the most variation in imped-ance values (76.4 %), while PC2 accounted for the next largest variation (23.6 %). Other classification algorithm such as K-nearest neighbors, decision trees, and support vector machines were also effective to carry out such tasks [37]. Distance of samples in sugar subsets were closer due to the larger molecular volume are difficult to adsorb and de-sorb under the AC current disturbance of EIS. Overall, the three subset could be distinguished clearly using our electrodes, giving an evaluation of the practical dish.
Point 11: The commercial IE consisted of Cu/Ni/Au with thickness of 12 μm, 1 μm, and 1 μm’ – There is no information about the electrode manufacturer. Do the details pertain to the thickness of the Cu (12 µm), Ni (1 µm), and Au (1 µm) layers? What was the composition of the electrode surface that was in contact with examined solution? Was it pure gold?
Response 11: ** Thanks for this valuable suggestion here. The surface layer of the IE electrode is gold, which was provided by the datasheet from the supplier. Here is revision in Section Materials and Methods “The IE electrodes consisted of Cu/Ni/Au were commercial available.”, and the revision in Section Results and Discussions“As illustrated in Fig. 1, the metal layer of commerical IE of 20 pairs fingers were consisted of Cu/Ni/Au with thickness of 12 μm, 1 μm, and 1 μm from bottom to top layers. Cu layer provided the conductivity, Ni layer was deposited for adhesion, and Au layer used to provide stable chemical interfaces.”
Point 12: 5 mg mL-1 GO was’ - Was it a water suspension?
Response 12: ** We thank the reviewer’s good comments here. It was a water suspension. The term was revised into “aqueous GO solution” to make it clear.
Point 13: GO solution was the electrolyte’, ‘And buffer solution was used to control the pH in case of gas generation’ - Please organise the information about the solutions used, namely, provide details about the composition of the solution used for depositing GO on the IE, including the concentration of GO, e.g. the optimal one. Was the IE electrode rinsed after depositing GO? If so, please provide information on what was used to rinse the electrode. Was more than one buffer used during the deposition of GO on the IE surface?
What was the pH of the solution used to deposit GO, described in the manuscript as 'The buffer solution was a mixture of NaCl (0.4 g L-1), KCl (0.4 g L-1), CaCl2·H2O (0.795 g L-1), urea (1 g L-1), Na2S·H2O (0.005 g L-1), NaH2PO4·H2O (0.78 g L-1)'? Was the composition of the solution taken from the literature, or is it the result of the authors' research? Please provide literary references. At what potential was the hydrogen evolution observed? Did the hydrogen evolution interfere with the deposition of GO?
Response 13: ** We thank the valuable comment here. Aqueous 5 mg mL-1 GO was diluted with buffer solution into 1 mg mL-1 GO contents was defined as the electrolytes. And the sentence “GO solution was the electrolyte’” was deleted in this revision. And the second sentence was revised into “And buffer solution was used to control the pH to suppress the gas generation”.
The solution and the composition used for depositing GO was provided in Section 2.2 Preparation of ERGO modified IE. We compared the concentration of GO of 5 mg mL-1 GO and 1 mg mL-1 GO, and choose the 1 mg mL-1 GO due to the better morphology after the deposition, Fig.3a-b described the electrochemical properties at different concentrations.
After depositing GO, the IE electrode was cleaned. And the process was added in the revision “After electrodeposition, IE was gently picked out and immersed in 100 ml DIW for 10 min. Then, gently rinsed by DIW and alcohol, and dried naturally.”
The composition of buffer solution was commercial available as the so called “artificial saliva” according to ISO/TR10271 standard (Anal. Methods, 2021, 13, 56-63). Literary references was added. During the deposition of GO, no hydrogen evolution at interfere was observed.
Point 14: 2.3 Evaluation method of modelled dish’ - The entire chapter needs to be revised, including the title of the chapter.
Provide information about the composition of the standard solutions and the solution used to record the EIS curves.
'Afterwards, 245 μL of NaCl solution was added into the model dish each time, for 20 times. After each addition, the corresponding EIS was recorded.' – What was the purpose of these studies? Was the EIS response recorded to obtain a calibration curve? The results for the 20 NaCl additions are not presented in the work.
Response 14: ** We thank the valuable comment here. The title of the chapter was revised into “Method for database establishment for dish evaluation”. And the entire chapter was revised. Modeled dishes: A mixed solution of 513 mmol L-1 sodium chloride (NaCl), 147 mmol L-1 sugar, and 175 mmol L-1 monosodium L-glutamate (MSG) was prepared ac-cording to the sauce of certain Chinese stir-fried dishes. Above mixed solution was used as the standard solution.
The step by step addition of NaCl solution was the data base collection process, which was used to for the PCA in the following process. All the data were presented in Figure 4a-c, and please refer to the corresponding discussions in Section 3.
Point 15: Results and Discussions The chapter should be divided into sub-sections: (i) Modification of Au electrodes with rGO (possible/optional subtitle for the subsection).
In this chapter, microscopic studies, XRD, Raman, and CA should be included. To allow the reader to assess the changes caused by the reduction of GO to rGO, please add the XRD curves recorded for GO before electroreduction.
What was the reduction potential used during the rGO study shown in Figures 2b and 2c?
Can the authors add to the figure the Raman spectra of rGO obtained after reduction at -1.1 V, -1.3 V, and -1.5 V? Did the wetting angle measured for the modified IE/rGO electrode surfaces depend on the reduction potential of GO? What reduction potential was used to prepare the electrodes for which CA was measured, as seen in Figure 2d?
What was the reduction potential used to create the rGO visible in Figures 2e-2f? Regarding 'Samples were scraped off and dispersed under ultrasonic' – How were the rGO samples prepared for XRD and Raman studies?
At the end of the chapter, information about the optimal conditions for preparation of rGO/Au electrodes should be included, considering the buffer composition, pH, GO concentration, reduction potential, and deposition time.
(ii) Use of rGO/Au electrodes to study solutions that simulate the composition of spices (possible/optional subtitle for the subsection). This chapter should contain information on EIS studies.
Response 15: ** We thank the valuable comment here.
Subtitles were added as follows: (i) Modification of IE electrodes with ERGO; (ii) Electrochemical characterization of individual ingredient. (iii) Classification-based evaluation in dimension of cook seasoning
Before the reduction, GO was amorphous. Almost no graphite lattice existed in GO. So during the XRD characterization, no information could obtained.
A direct current voltage of -1 V was applied for 300 s during the GO electrodeposition as shown in 2.2 Preparation of ERGO modified IE.
The sample used for CA measured used the method described in Section 2.2 Preparation of ERGO modified IE. The electrodeposition of GO is a pretty mature technology. And many reference could be found. The related characterization data is abundant. We paid more attention on the application of the modified electrodes. And the discussion and interpretation of EIS curves were provided when describing the Figures.
The optimal conditions for preparation of rGO/Au electrodes including buffer composition, pH, GO concentration, reduction potential, and deposition time were including in Section 2.2 Preparation of ERGO modified IE. The sample used for XRD and Raman measurement also used the method described in Section 2.2 Preparation of ERGO modified IE.
We believe that refer to our method, other researchers could replicate the same experimental results.
Point 16: .(a-b) GO of different concentration showed different EIS’ - This is a conclusion, change the figure description. There is no information about the conditions for preparing the electrode and the composition of the solutions studied. From the figure, it is not clear whether single-component or multicomponent solutions were examined.
Response 16: ** We thank the valuable comment here. GO solution for deposition is multicomponent solutions. Aqueous GO 5 mg mL-1 GO was diluted with buffer solution into 1 mg mL-1. As described in the above Section 2.2 Preparation of ERGO modified IE, “The buffer solution was a mixture of NaCl (0.4 g L-1), KCl (0.4 g L-1), CaCl2·H2O (0.795 g L-1), urea (1 g L-1), Na2S·H2O (0.005 g L-1), NaH2PO4·H2O (0.78 g L-1).”
The description of the figures were revised in “The Nyquist curves (a) and the Bode curves (a) of GO solution of 1mg mL-1 and 5mg mL-1 used for ERGO deposition.”
Point 17: EIS behavior of three pure components have also been studied to demonstrate the data feature’ - Please revise; the sentence is unclear.
Response 17: ** Thanks for this valuable suggestion here. This sentence is unclear and misleading, and was deleted.
Point 18: the Nyquist curves showed distinct differences’ - Information about Rs for solutions of different compounds and concentrations should be placed in a table.
Response 18: ** Thanks for this valuable suggestion here. The information of Rs for solutions with different compounds were rearranged into a table as follows. And the corresponding revision was in the manuscript.
Table 1. Rs values in EIS curves of the samples in three subsets obtained
|
Categories |
Concentrations of subsets |
|||
|
1 mM (Ω) |
10 mM (Ω) |
100 mM (Ω) |
1000 mM (Ω) |
|
|
NaCl |
688.5 |
91 |
25 |
7.5 |
|
MSG |
631.1 |
71.7 |
7.4 |
1.8 |
|
Sugar |
23278 |
19875 |
16434 |
779 |
Point 19: We refer to the concentration certain Chinese stir-fired dishes, and diluted for 10 times as pretreatment’ - Please revise; the sentence is unclear.
Response 19: ** We thank the valuable suggestion here. This sentence was misleading and redundant, and was deleted in this revision.
Point 20: we developed an evaluation system for dishes’ - This conclusion goes too far. The authors investigated only a synthetic solution consisting of NaCl, sugar, and monosodium L-glutamate. Such a solution can at best be an approximation of the composition of a seasoning in liquid form, not a finished dish.
Response 20: ** We thank the valuable comment here. The mixed solution is an approximation of the composition of a seasoning. In this revision, we supplemented the data of an actual dish as shown in Fig. 4(d-e). Even though, we treated this method as a preliminary study. The conclusion was revised into “we developed a preliminary method for mixed solution classification, which could be potentially used in dish evaluation.” And “Further, we demonstrated the effectiveness of this method on scrambled eggs with tomato. Three subsets were classified with PCA to give show which cooking seasoning has been added. We believe this work might have potential application in evaluation of dishes or environmental monitoring with mixed solution system.”
Point 21: Please correct the description of Figure 1. The use of the word 'illustrator' is not appropriate.
** We thank the valuable comment here. The word 'illustrator' is revised into “Schematic diagram” here.

Round 2
Reviewer 1 Report
The author answered all the questions well and made modifications to the article
The English in the manuscript can express the author's meaning.
Author Response
We appreciate the comments from dear referee, helping us improve the quality. For the purpose of clarity, the point-to-point answers are marked with RED color and started with “**” below. And please see the attachment of answers.
Point 1: The author answered all the questions well and made modifications to the article.

Reviewer 2 Report
The authors have considered most of my comments; however, the manuscript still requires some corrections.
Lines 33 – 35: Two sentences with identical meanings. One of them should be eliminated.
Line 46: Replace ‘potentiometer’ with ‘such as potentiometry’
Line 87: ‘The IE electrodes consisted of Cu/Ni/Au were commercial available’ – Replace ‘commercial available’ with ‘commercially available’ and provide the details about their manufacturer.
Line 102: add ‘dispersion’ or its synonym in the sentence ‘Aqueous 5 mg mL-1 GO dispersion was diluted…’ and elaborate the sentence to be more informative. The example is provided below.
‘A 5 mg / ml dispersion of GO was diluted with buffer solution to produce a concentration of 1 mg / mL of GO, which was then used as an electrolyte in subsequent studies’ or synonym.
Lines 114, 117, 121, and the rest of the manuscript: The term ‘sugar’ is too ambiguous; please use the chemical name that indicates sucrose.
Figure 2: There is still no information on the conditions under which the GO was reduced to obtain the rGO presented in the figure.
Table 1 - The text lacks a reference to Table 1. The information provided in lines 219 to 223 is a repetition of the content presented in Table 1 and no commentary was provided.
Author Response
We appreciate the comments from dear referee, helping us improve the quality. For the purpose of clarity, the point-to-point answers are marked with RED color and started with “**” below. And please see the attachment of answers with figures.
Point 1: The authors have considered most of my comments; however, the manuscript still requires some corrections.
Response 1: ** We thank the reviewer’s highly positive and valuable comment. And revisions have been made. Please check it.
Point 2: Lines 33 – 35: Two sentences with identical meanings. One of them should be eliminated.
Response 2: ** Thanks for this valuable suggestion here. . The sentence “Taste is hopeful to realize automation because it is fundamentally a chemical process” has been eliminated, and the reference has been moved to the previous sentence.
Point 3: Line 46: Replace ‘potentiometer’ with ‘such as potentiometry’
Response 3: ** We thank the reviewer’s good comments here. Your suggestions is more accurate and appropriate here. The word “potentiometer” was replaced with “potentiometry”.
Point 4: Line 87: The IE electrodes consisted of Cu/Ni/Au were commercial available’ – Replace ‘commercial available’ with ‘commercially available’ and provide the details about their manufacturer.
Response 4: ** We thank the valuable comment here. The typo was revised. And the manufacturer, Changchun Beirun Electronic Technology Co., Ltd was added in the revision.
Point 5: Line 102: add ‘dispersion’ or its synonym in the sentence ‘Aqueous 5 mg mL-1 GO dispersion was diluted…’ and elaborate the sentence to be more informative. The example is provided below. ‘A 5 mg / ml dispersion of GO was diluted with buffer solution to produce a concentration of 1 mg / mL of GO, which was then used as an electrolyte in subsequent studies’ or synonym.
Response 5:. ** We thank the valuable comment here. According your kind suggestion. The sentence was revised into “A 5 mg mL-1 GO aqueous dispersion was diluted with buffer solution to produce a concentration of 1 mg mL-1 GO dispersion, which was then used as an electrolyte for the electrochemical deposition.” to be more informative.
Point 6: Lines 114, 117, 121, and the rest of the manuscript: The term ‘sugar’ is too ambiguous; please use the chemical name that indicates sucrose.
Response 6: ** We thank the valuable comment here. Sugar was replace with sucrose throughout the entire manuscript. And the legends in the figures were also revised. Figures 3,4
Point 7: Figure 2: There is still no information on the conditions under which the GO was reduced to obtain the rGO presented in the figure.
Response 7: ** We thank the valuable suggestion here. The condition of GO reduction was provided in the Paragraph 2, Section 2.1 Materials and characterization as “A direct current voltage of -1 V was applied for 300 s. Meantime, we add the condition of GO reduction on the legend to presented the condition in the figures of material characterization.
Point 8: Table 1. The text lacks a reference to Table 1. The information provided in lines 219 to 223 is a repetition of the content presented in Table 1 and no commentary was provided.
Response 8: ** Thanks for this valuable suggestion here. Table 1 was referred in the text, and the repeated lines 219 to 223 was replaced with commentary, as follows: “The equivalent series resistances, Rs, were summarized in Table 1. All the conductivity was decreased as the concentration increased. At certain concentration, the order of conductivity was MSG, NaCl, and Sucrose. Even sucrose was molecular crystal without conductivity, EIS could provide the Rs results, which might be the interaction at the interfaces due to the molecular polarity.”
